# Bibliometric Analysis of Forestry Research in Mexico Published by Mexican Journals

Alberto Santillán-Fernández [1,*], Nehemias Vásquez-Bautista [2], Luis Marcelino Pelcastre-Ruiz [3], Carlos Antonio Ortigoza-García [3], Edgar Padilla-Herrera [3], Alfredo Esteban Tadeo-Noble [1], Eugenio Carrillo-Ávila [3], José Francisco Juárez-López [4], Javier Enrique Vera-López [3] and Jaime Bautista-Ortega [3]

1. Catedrático-Conacyt, Colegio de Postgraduados Campus Campeche, Champotón 24450, Campeche, Mexico
2. Fideicomisos Instituidos en Relación con la Agricultura, Tuxtla Gutiérrez 29030, Chiapas, Mexico
3. Postgraduate in Bioprospecting and Agricultural Sustainability in the Tropics, Colegio de Postgraduados Campus Campeche, Champotón 24450, Campeche, Mexico
4. Colegio de Postgraduados Campus Tabasco, Cárdenas 86500, Tabasco, Mexico
* Correspondence: santillan.alberto@colpos.mx; Tel.: +52-22-21-0-77-329

**Abstract:** There is scarce research assessing the productivity of scientific articles on forestry topics. The objective of this study was to analyze the scientific production on forestry topics that originated in Mexico and were published in Mexican journals from 1996 to 2019 and to identify the causes that determine the impact factor of such publications and the space-time evolution of forestry research in Mexico. In addition, to analyze whether researchers tend to publish in journals published by their affiliation institutions. The study considered 2384 scientific articles from seven journals belonging to category VI of Biotechnology and Agricultural Sciences listed in the Journals Classification System by the National Council of Science and Technology that publishes forestry topics. Bibliometric indicators were generated through text mining and analysis of co-authorship networks. It was found that forestry research in Mexico from 1996 to 2019 presented exponential growth in the number of publications. Forestry scientific production was concentrated in the center of the country. It was dominated by researchers from three of 122 institutions: Instituto Nacional de Investigaciones Forestales, Agrícolas y Pecuarias (13.88%), Colegio de Postgraduados (12.50%), and Universidad Autonoma Chapingo (10.44%). The journals with the highest number of publications were: Revista Mexicana de Ciencias Forestales (26.51%), Revista Chapingo Serie Ciencias Forestales y del Ambiente (20.34%), and Madera y Bosques (18.88%). Results show that forestry researchers in Mexico published mostly in journals edited by their affiliation institutions, which restricts constructive criticism of peer review and increases academic endogamy. Also showed the need to generate more forestry research for the southeast of the country on topics such as climate change, carbon capture, forest biometry, and remote perception, which are relevant aspects when we consider that no published research evaluated the development of the forestry sector in Mexico.

**Keywords:** academic endogamy; forest; forestry scientific journals; scientific article; scientific trends

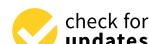



## 1. Introduction

The importance of the forestry sector in Mexico has been analyzed from several approaches, as a sector that contributes 0.99% of the National Gross Domestic Product [1], as a supplier of ecosystem services [2,3], as an agent of community development [4,5], and even from an ethnographic approach [6]. However, up to now, there is scarce research assessing the productivity of scientific articles on forestry topics [7].

The publication of a scientific study is the most effective way of transmitting knowledge acquired as a consequence of the research, and its visibility is important for the researchers themselves, for their affiliation institutions, and for the organizations that

finance the research [8]. Increased scientific production in recent decades and its indexation in automatized bibliographic databases have potentiated the use of bibliometry, most likely because bibliometric analyses enable the generation of indicators to measure the results from scientific and technological activities [9–11].

Bibliometric studies from published scientific articles allow generating indicators and mathematical models to characterize the development and evolution of the frequency and quality of the publications [12,13]. In Mexico, one of the first bibliometric studies was the one by Peña [14], who analyzed the state of national scientific research and found that the low salaries of the academic sector and the null dissemination of information until that moment had hindered an increase in publications, despite the creation of the National System of Researchers (Sistema Nacional de Investigadores, SNI) in 1984.

Bibliometric studies have been conducted in the forestry sector for specific themes, such as silviculture [15], community forestry development [16], the use of drones in the determination of forest biomass [17], and even to evaluate national forestry systems, as in the case of India [18] and Bangladesh [19]. Furthermore, to evaluate the scientific productivity of forestry researchers in Tanzania [20] and India [21] and high-impact journals in the Journal Citation Reports such as Forests [22].

In Mexico, bibliometric techniques have been used to analyze forestry biometry for the integral management of the forests [7,23]; growth rings for the potential estimation of carbon [24]; the use of drones for forest management [25]; and to identify research needs on the carbon and nitrogen dynamics in agroforestry systems [26]. However, these studies analyze a specific topic from the forestry sector and are not focused on evaluating the sector as a whole.

Scientific journals in Mexico are growing. They have mostly been created by institutions to publish results from studies performed within their academic programs or the projects that each directs [27]. In the forestry sector, institutions, such as Universidad Autónoma Chapingo, Colegio de Postgraduados, and Instituto Nacional de Investigaciones Forestales, Agrícolas y Pecuarias, have contributed to the development of the sector through the publication of scientific and technological research in their respective journals [28].

The most studied topics of forestry research in Mexico are focused on biometry, silviculture, plantations, nurseries, and ecosystem services, and exponential growth since 2000 [7]. However, despite this growth in the number of publications, there is scarce research to assess the evolution and the impact of these publications [7]. The objective of this study was to analyze the scientific production on forestry topics that originated in Mexico and were published in Mexican journals from 1996 to 2019 to identify the causes that determine the impact factor of such publications and the space-time evolution of forestry research in Mexico. In addition, to analyze whether researchers tend to publish in journals published by their affiliation institutions.

## 2. Materials and Methods

### 2.1. Origin of the Information and Data Preparation

This study considered the scientific articles available in the journals to category VI of Biotechnology and Agricultural Sciences listed in the Classification System of Mexican Journals of the National Council of Science and Technology, which publishes forestry topics, according to the guide for authors to the journals [29]. The scientific articles available on their web pages from 1996 to 2019 (a 24-year period) were compiled. Only the articles whose information originated in Mexico were considered; that is, studies from foreign authors who conducted studies in Mexico were included, and studies whose study area was not located in Mexico were excluded even when published by a Mexican journal.

The variables analyzed from each of the articles were: Name of the Journal to evaluate which journals have published most frequently from 1996 to 2019; First Author and Collaborators, serving to aid in understanding the network of actors involved in forestry research in the country; Year, to locate the information in a timeline; Institution, to evaluate

the frequency of publications from the institutions that perform forestry research, and in the case of institutions with many campuses, each one was considered.

The Postal code of the first author's institution was also included. To carry out a geographic location of the institution where the information originated (in cases where the postal address was not reported), the institution's name was found with Google Earth® tools. In the official pages of the institutions, for the case where the first author was not from a Mexican institution, the corresponding author's affiliation institution was used.

Finally, the Title, Abstract, and Keywords were used to categorize the forestry topic addressed by the publication; experts in forestry sciences from Universidad Autónoma Chapingo [2] and Colegio de Postgraduados campus Montecillo [2] were consulted for the definition of the topics. The experts in forest science were selected based on their availability, references among the scientific community, and their permanence in the National System of Researchers of Conacyt for more than 10 years. The frequency of publication in the period of analysis and its mean annual growth rate (MAGR, %) was determined for each topic. The mathematical expression was the following:

$$MAGR_{topic} = \left( \frac{A}{B} \right)^{\frac{1}{N}} - 1$$

where:

*A* = Number of scientific articles registered in the final year;
*B* = Number of scientific articles registered in the initial year;
*N* = Period years old from B to A.

Variables entry was done in a spreadsheet, and the language of all the texts analyzed was Spanish, the original language in which the analyzed journals were published. Some records were standardized while data entry all the information since the information available in the articles was sometimes incomplete or presented with variations. In addition, to facilitate the analysis, special characters were eliminated or changed, such as ñ (by n), accents, superindices, subindices, ®, ©, among others [30]. For interpretation purposes, the information was separated into three periods: 1996–2000, 2001–2010, and 2011–2019.

### 2.2. Text Mining Analysis

The frequency of articles per year, journal, the institution where the research originated, articles where the principal author was from the same institution as the journal, and the topic corresponding to each article, were obtained through the complement RcmdrPlugin.temis of the statistical software R [31]. The texts published on the different campuses of an institution were considered as part of the main institution. For the case of the frequencies per year and journal, a least-squares regression model was adjusted, which allowed defining of the temporal behavior of forestry scientific production from 1996 to 2019 [32].

### 2.3. Network Analysis

To show the evolution in the number of institutions and authors that published forestry texts in journals edited in Mexico from 1996 to 2019, a network of institutions and authors was constructed with the help of Ghephi software [33]. The texts published on different campuses of an institution are considered to be part of the main institution. The analysis periods were: 1996–2000, 2001–2010, and 2011–2019.

### 2.4. Spatial Representation of the Institutions

With the help of the Postal Code variable and Google Earth® tools, the geographic coordinates were obtained (Longitude, Latitude) from the institution of the first author of each one of the articles published from 1996 to 2019. The spatial representation of the number of articles per institution was made with the geographic package ArcGIS® [34]. In institutions with several campuses, each was considered. The spatial representation was complemented with georeferenced areas where there are forests and rainforests in Mexico,

based on Series VI of Land Use and Vegetation of the National Institute of Statistics and Geography [35].

## 3. Results and Discussion

In category VI of the Biotechnology and Agricultural Sciences in the Classification System of Mexican Science and Technology Journals [29], there are 24 scientific journals, and, according to the guideline for authors to the journals, seven journals published topics related to the forestry sector. The specialized journals are: Madera y Bosques (MyB), Revista Chapingo Serie Ciencias Forestales y del Ambiente (RCSCFA), and Revista Mexicana de Ciencias Forestales (RMCF); the journals Polibotánica (PB), Agrociencia (AG), Ecosistemas y Recursos Agropecuarios (ERA), and Tropical and Subtropical Ecosystems (TSE) are multidisciplinary in botanical, environmental and production system topics.

Table 1 shows the description and impact factors of the seven journals which published forestry scientific texts from 1996 to 2019 in Mexico. The highest number of texts were published in specialized journals. RMCF concentrated the largest number, with 632 texts, although it also presented the lowest impact factor since it is classified as a journal in development [29]. The low impact level of Mexican journals has been documented by López-Leyva [36], who found that elements, such as the publication language (Spanish) and the priority in publishing studies whose authors belong to the same institution that edits the scientific journals, reduce the visibility of the publications.

**Table 1.** Description and impact factors of journals that publish forestry topics in Mexico based on the journal classification system by Conacyt [26].

| Journal | | | | Impact Factor * | | |
|---------|-------------|----------|-----------|-------|-------------|--------------------------|
| Name | Institution | Articles | Period | JCR | SCOPUS/SJR | Conacyt |
| Madera y Bosques (MyB) | Instituto de Ecología, A.C. | 450 | 1996–2019 | 0.583 | 0.260 | Q3 |
| Agrociencia (AG) | Colegio de Postgraduados | 220 | 1996–2019 | 0.370 | 0.182 | Q3 |
| Revista Chapingo Serie Ciencias Forestales y del Ambiente (RCSCFA) | Universidad Autónoma Chapingo | 485 | 1998 -2019 | 0.554 | 0.211 | Q3 |
| Polibotánica (PB) | IPN | 370 | 1996–2019 | — | — | International competence |
| Revista Mexicana de Ciencias Forestales (RMCF) | INIFAP | 632 | 1996–2019 | — | — | In development |
| Ecosistemas y Recursos Agropecuarios (ERA) | Universidad Juárez Autónoma de Tabasco | 132 | 1996–2019 | — | — | Internacional competence |
| Tropical and Subtropical Ecosystems (TSE) | Universidad Autónoma de Yucatán | 95 | 2009–2019 | — | 0.148 | Q4 |

* Indicators obtained from the Classification System of Mexican Journals of Science and Technology: VI. Biotechnology and Agricultural Sciences. Those that appear in the journal's portal were not considered. For the case of RMCF, its website indicates an Impact Factor 2019 = 0.2097 SciELO Citation Index–Web of Science. IPN = Instituto Politécnico Nacional; INIFAP = Instituto Nacional de Investigaciones Forestales, Agrícolas y Pecuarias.

From 1996 to 2019, 2384 articles were published in Mexico of purely forestry studies; 257 corresponded to the period 1996–2000, 705 to 2001–2010, and 1422 to 2011–2019; 65.73% of the studies published were focused on three of the seven journals analyzed: RMCF (26.51%), RCSCFA (20.34%), and MyB (18.88%). An exponential trend was seen in the increase of publications ($R^2 = 0.8864$) (Figure 1). Gómez and Hernández [37] and Martínez-

Santiago et al. [7] attribute this growth to the policies that Conacyt implemented in 1984 with the creation of the National Researchers System (Sistema Nacional de Investigadores, SNI) and the admission of more master and doctoral degrees to the PNPC program of Conacyt (Padrón Nacional de Posgrados de Calidad). However, Bravo-Vinaja and Sanz-Casado [38] and López-Leyva [36] consider that the increase in the number of publications Mexico has produced since the creation of the SNI has only been in quantity and not in quality, which is reflected in the low impact of Mexican journals.

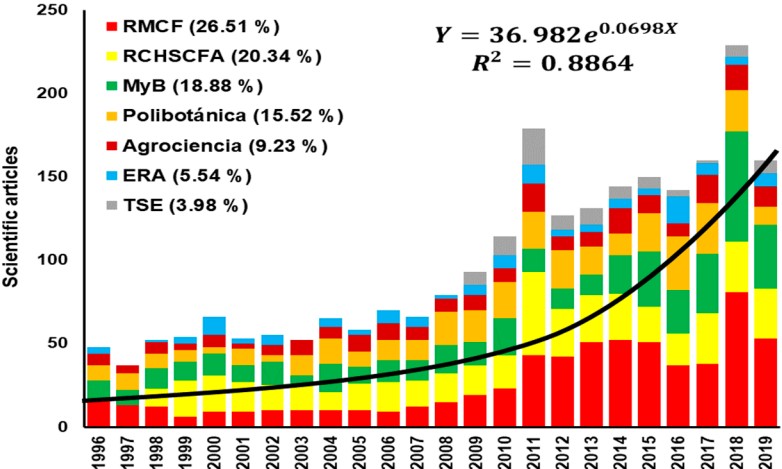

**Figure 1.** Time evolution per journal of the forestry publications in Mexico from 1996 to 2019.

From the 2384 scientific articles found for 1996–2019, the highest number corresponded to themes related to the sustainable management of natural resources (281 articles, 11.79%), forest ecology (201, 8.43%), forest botany (190, 7.97%), and dasometry (167, 7.01%). From 1996–2000, 17 topics were counted, 20 from 2001–2010, and 27 from 2011–2019 (Table 2). The diversification of the topics is explained by Chao et al. [39] and Martínez-Santiago et al. [7] because the development of technology has allowed the creation of specialized software that broadens the action field of science, such as remote perception, biometry, and estimation of carbon capture.

**Table 2.** Frequency and mean annual growth rate per forestry topic of the scientific articles published by Mexican journals from 1996 to 2019.

| Topic | 1996–2000 | | 2001–2010 | | 2011–2019 | | 1996–2019 | | MAGR(%) |
|---|---|---|---|---|---|---|---|---|---|
| | Number | % | Number | % | Number | % | Number | % | |
| Forest botany | 45 | 17.51 | 73 | 10.35 | 72 | 5.06 | 190 | 7.97 | 2.06 |
| Dasometry | 33 | 12.84 | 55 | 7.80 | 79 | 5.56 | 167 | 7.01 | 3.87 |
| Wood anatomy | 30 | 11.67 | 39 | 5.53 | 37 | 2.60 | 106 | 4.45 | 0.92 |
| Sustainable management of natural resources | 28 | 10.89 | 97 | 13.76 | 156 | 10.97 | 281 | 11.79 | 7.75 |
| Forest ecology | 22 | 8.56 | 52 | 7.38 | 127 | 8.93 | 201 | 8.43 | 7.92 |
| Forest nursery | 19 | 7.39 | 41 | 5.82 | 58 | 4.08 | 118 | 4.95 | 4.97 |
| Forest health | 14 | 5.45 | 51 | 7.23 | 70 | 4.92 | 135 | 5.66 | 7.25 |
| Silviculture | 13 | 5.06 | 54 | 7.66 | 25 | 1.76 | 92 | 3.86 | 2.88 |
| Conservation and restoration of natural resources | 12 | 4.67 | 52 | 7.38 | 66 | 4.64 | 130 | 5.45 | 7.69 |
| Plant breeding and forest germplasm | 11 | 4.28 | 30 | 4.26 | 65 | 4.57 | 106 | 4.45 | 8.03 |
| Forest industries | 10 | 3.89 | 19 | 2.70 | 14 | 0.98 | 43 | 1.80 | 1.47 |

**Table 2.** *Cont.*

| Topic | 1996–2000 | | 2001–2010 | | 2011–2019 | | 1996–2019 | | MAGR(%) |
|---|---|---|---|---|---|---|---|---|---|
| | Number | % | Number | % | Number | % | Number | % | |
| Forest soils | 7 | 2.72 | 9 | 1.28 | 14 | 0.98 | 30 | 1.26 | 3.06 |
| Climate change | 2 | 0.78 | 29 | 4.11 | 95 | 6.68 | 126 | 5.29 | 18.28 |
| Economy and forest legislation | 4 | 1.56 | 23 | 3.26 | 56 | 3.94 | 83 | 3.48 | 12.16 |
| Forest biometry | 3 | 1.17 | 13 | 1.84 | 104 | 7.31 | 120 | 5.03 | 16.67 |
| Fire management | 2 | 0.78 | 28 | 3.97 | 30 | 2.11 | 60 | 2.52 | 12.50 |
| Agrosilvopastoral | 2 | 0.78 | 13 | 1.84 | 19 | 1.34 | 34 | 1.43 | 10.28 |
| Fauna | 0 | 0.00 | 9 | 1.28 | 11 | 0.77 | 20 | 0.84 | 1.12 |
| Remote perception | 0 | 0.00 | 13 | 1.84 | 124 | 8.72 | 137 | 5.75 | 13.35 |
| Carbon capture | 0 | 0.00 | 5 | 0.71 | 92 | 6.47 | 97 | 4.07 | 17.56 |
| Others * | 0 | 0.00 | 0 | 0.00 | 108 | 7.59 | 108 | 4.53 | 3.42 |
| Total | 257 | 100.00 | 705 | 100.00 | 1422 | 100.00 | 2384 | 100.00 | |

* Includes topics related to bio-prospection, biopolymers, ecotourism, bibliometrics, biogeochemistry, bioethanol, and electricity.

Table 2 shows that for the period 1996–2000, the topics with the highest frequency of publications were forest botany (17.51%) and dasometry (12.84%); in the period 2001–2010, sustainable management of natural resources (13.76%) and forest botany (10.35%); and in 2011–2019, sustainable management of natural resources (10.97%) and forest ecology (8.93%). However, the topics with the highest mean annual growth rate (MAGR) were climate change (18.28%), carbon capture (17.56%), forest biometry (16.67%), and remote perception (13.35%).

The variety of topics per analysis period is explained based on Santillán-Fernández et al. [40] because the researchers defined their action fields in the function of the scientific trends in agreement with current international and national themes. For the period 2011–2019, the trending themes in forestry sciences were: climate change [41], carbon capture [26], forestry biometry [23], and remote perception [25].

### 3.1. Evolution of the Network of Authors Who Published Forestry Research in Mexico

From the 2384 articles analyzed, 1603 different first authors were found. Between the author and collaborators, there were a total of 3798 different individuals. Figure 2 shows an increase in the number of authors (nodes) and connections between authors of the network (connections) for the period analyzed. However, the density of the co-authorship network has decreased, and the interactions in the network results are very weak. The density is an indicator of the Social Network Analysis (SNA) implying that the nodes interact (are connected) with one another; mathematically, it is a value within the 0–1 interval, and the closer to 1 the interaction, the greater the network [42].

The low interaction between the authors who published forestry texts from 1996 to 2019 has been documented by Martínez-Santiago et al. [7], who found that researchers prefer to form groups inside their university and prioritize personal affinities rather than seek collaborations with researchers from other institutions. This explains why there were articles with a maximum of 4 authors for 1996–2000, a maximum of 6 authors for 2001–2010, and up to 9 authors from the same institution for 2011–2019. According to López-Leyva [36], forming intra-university research groups limits the impact of the study developed by restricting constructive criticism.

The publication of forestry articles in journals edited by the first author's affiliation institution were 29.53% of the total texts (704 out of 2384); 53.70% (138 out of 257) from 1996–2000, 38.58% (272 out of 705) from 2001–2010, and 20.68% (294 out of 1422) from 2011–2019. The journals that published most texts from institutional authors were AG from Colegio de Postgraduados (40.45%); RMCF from Instituto Nacional de Investigaciones Forestales, Agrícolas y Pecuarias (38.77%); and RCSCFA from Universidad Autónoma Chapingo (33.81%) (Table 3). However, in contrast to RMCF and RCSCFA, specialized

journals for forestry themes, AG is a multidisciplinary journal in botanical, environmental, and production systems topics.

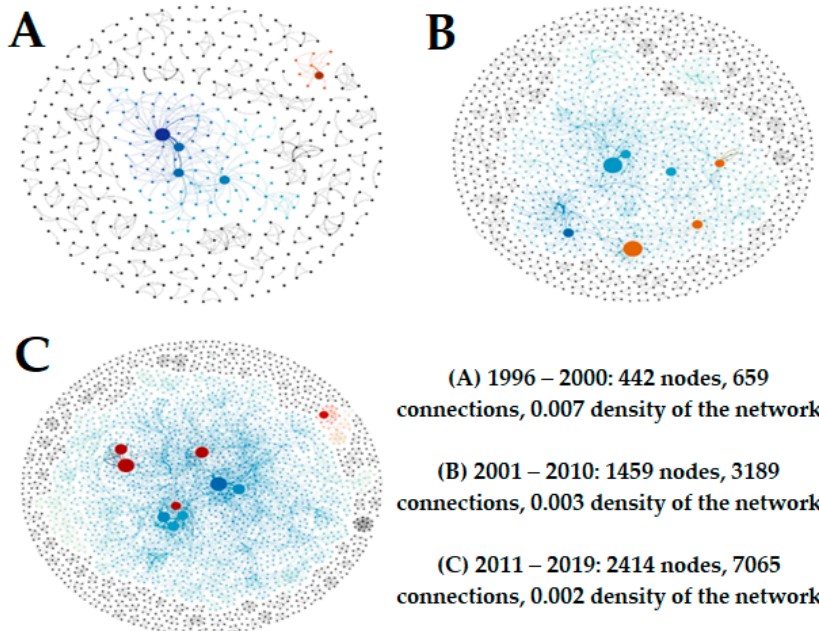

(A) 1996 – 2000: 442 nodes, 659 connections, 0.007 density of the network

(B) 2001 – 2010: 1459 nodes, 3189 connections, 0.003 density of the network

(C) 2011 – 2019: 2414 nodes, 7065 connections, 0.002 density of the network

**Figure 2.** Networks of authors and collaborators per period of analysis. Each node refers to one author, and the node's size corresponds to the number of texts where they appear as the author. The connections refer to connections between authors.

**Table 3.** Percentage of scientific articles per journal (SA) whose main author (MA) is from the same institution that edits the journal and which were published in Mexico from 1996 to 2019 with forestry topics.

| Journal * | 1996–2000 | | | 2001–2010 | | | 2011–2019 | | | 1996–2019 | | |
|---|---|---|---|---|---|---|---|---|---|---|---|---|
| | SA | MA | % | SA | MA | % | SA | MA | % | SA | MA | % |
| AG | 30 | 7 | 23.33 | 78 | 38 | 48.72 | 112 | 44 | 39.71 | 220 | 89 | 40.45 |
| ERA | 20 | 9 | 45.00 | 47 | 14 | 29.79 | 65 | 9 | 13.33 | 132 | 32 | 24.24 |
| MyB | 56 | 23 | 41.07 | 134 | 13 | 9.70 | 260 | 22 | 8.33 | 450 | 58 | 12.89 |
| PB | 39 | 20 | 51.28 | 135 | 59 | 43.70 | 196 | 29 | 14.62 | 370 | 108 | 29.19 |
| RCSCFA | 55 | 40 | 72.73 | 165 | 74 | 44.85 | 265 | 50 | 18.86 | 485 | 164 | 33.81 |
| RMCF | 57 | 39 | 68.42 | 127 | 70 | 55.12 | 448 | 136 | 30.43 | 632 | 245 | 38.77 |
| TSE | | | | 19 | 4 | 21.05 | 76 | 4 | 5.08 | 95 | 8 | 8.42 |
| Total | 257 | 138 | 53.70 | 705 | 272 | 38.58 | 1422 | 294 | 20.68 | 2384 | 704 | 29.53 |

* AG: Agrociencia; ERA: Ecosistemas y Recursos Agropecuarios; MyB: Madera y Bosques; PB: Polibotánica; RCSCFA: Revista Chapingo Serie Ciencias Forestales y del Ambiente; RMCF: Revista Mexicana de Ciencias Forestales; TSE: Tropical and Subtropical Ecosystems.

Table 3 shows that, in the case of the journals specialized in forestry themes (MyB, RMCF, and RCSCFA), they decreased the publication of texts by institutional authors in the analysis periods. However, of the three journals, only MyB and RCSCFA achieved impact factors in JCR of 0.583 and 0.554 for 2011–2019, respectively [29], which aligned with a greater reduction in the publication of texts from institutional authors. According to López-Leyva [36] and Osuna-Flores et al. [27], the publication of texts in journals edited by the same institution of the author in Mexico prevents the journals from reaching relevant impact factors by limiting peer review, a phenomenon that Aguado-López et al. [43] and Silva et al. [44] call academic endogamy and which consists in authors, editors, and arbiters belonging to the same network of collaboration and research.

*3.2. Evolution of the Network of Institutions That Published Forestry Research in Mexico*

From 1996 to 2019, there were a total of 473 institutions that appeared as adscription sites of the authors who published forestry texts in Mexican journals. By taking as reference the institution of the first author, where according to Sanjuanelo et al. [45] the research is developed, it was found that from 1996–2000 there were a total of 34 different institutions; from 2001–2010 it doubled to 74, and from 2011–2019 it tripled to 122. However, only three institutions published a significant amount of texts in each of the analysis periods: Universidad Autónoma Chapingo (Uach), Instituto Nacional de Investigaciones Forestales, Agrícolas y Pecuarias (Inifap), and Colegio de Postgraduados (Colpos). This is because, according to Conacyt [28], Uach, Inifap, and Colpos are the institutions with the largest number of registered forestry researchers. (Figure 3).

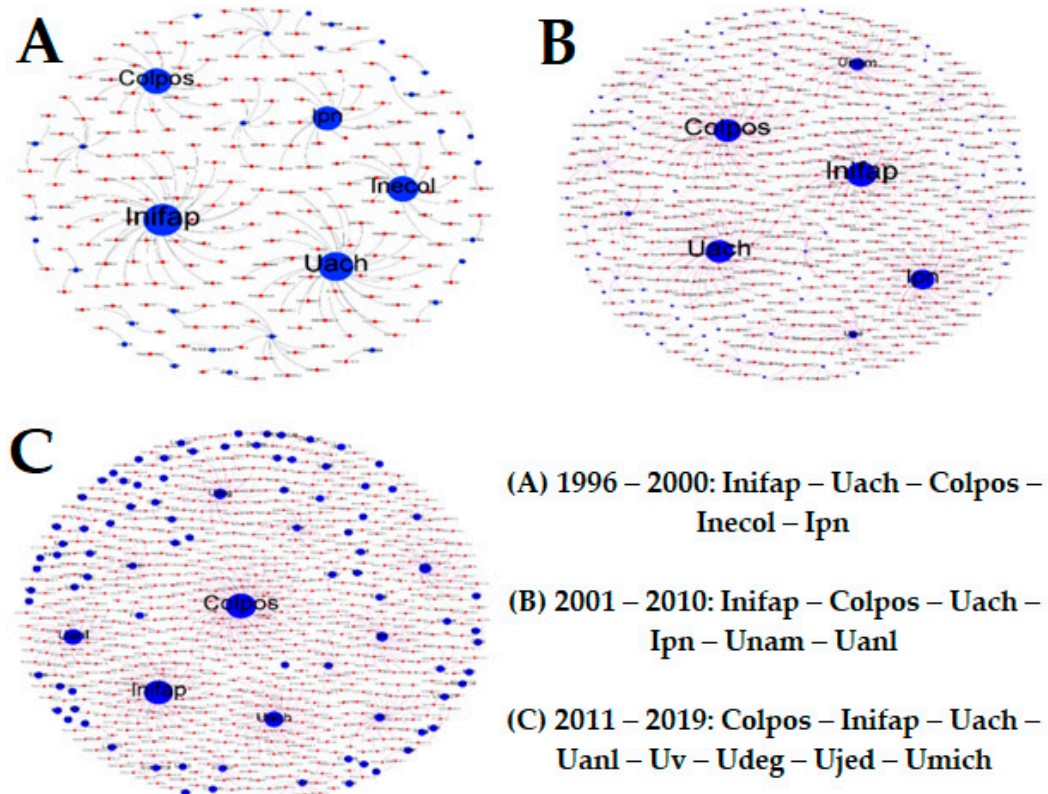

(A) 1996 – 2000: Inifap – Uach – Colpos – Inecol – Ipn

(B) 2001 – 2010: Inifap – Colpos – Uach – Ipn – Unam – Uanl

(C) 2011 – 2019: Colpos – Inifap – Uach – Uanl – Uv – Udeg – Ujed – Umich

**Figure 3.** Networks of institutions per period of analysis. Each node refers to one institution, and the node's size is a function of the number of texts where the institution appears referenced as the adscription site of an author.

Table 4 shows that the publication of forestry texts in Mexican journals for the period 1996–2000 was dominated in the hierarchical order by the number of texts by Inifap, Uach, and Colpos; for the period 2001–2010 the order was Inifap, Colpos, and Uach; and for the period 2011–2019, it was Colpos, Inifap, and Uach. The increase in the publications from Colpos since 2000 is a product of the repatriation of researchers trained outside the country at the beginning of the century. They have been gradually incorporated into the National Researchers System [7]. Uach (founded in 1854), Colpos (1959), and Inifap (1985) were created by the development of agricultural, forestry, and livestock production research in Mexico [4].

**Table 4.** Principal institutions that published scientific articles in Mexican journals about forestry topics from 1996 to 2019.

| Institution * | 1996–2000 | | 2001–2010 | | 2011–2019 | | 1996–2019 | |
|---|---|---|---|---|---|---|---|---|
| | Number | % | Number | % | Number | % | Number | % |
| Colpos | 28 | 10.89 | 91 | 12.91 | 179 | 12.59 | 298 | 12.50 |
| Inifap | 53 | 20.62 | 106 | 15.04 | 172 | 12.10 | 331 | 13.88 |
| Uach | 48 | 18.68 | 90 | 12.77 | 111 | 7.81 | 249 | 10.44 |
| Inecol | 25 | 9.73 | 20 | 2.84 | 14 | 0.98 | 59 | 2.47 |
| Ipn | 23 | 8.95 | 80 | 11.35 | 31 | 2.18 | 134 | 5.62 |
| Unam | 11 | 4.28 | 55 | 7.80 | 33 | 2.32 | 99 | 4.15 |
| Uanl | 10 | 3.89 | 28 | 3.97 | 98 | 6.89 | 136 | 5.70 |
| Uv | 2 | 0.78 | 20 | 2.84 | 62 | 4.36 | 84 | 3.52 |
| Udeg | 3 | 1.17 | 12 | 1.70 | 59 | 4.15 | 74 | 3.10 |
| Ujed | 0 | 0.00 | 6 | 0.85 | 41 | 2.88 | 47 | 1.97 |
| Umich | 3 | 1.17 | 9 | 1.28 | 39 | 2.74 | 51 | 2.14 |
| Otras | 51 | 19.84 | 188 | 26.67 | 583 | 41.00 | 822 | 34.48 |
| Total | 257 | 100.00 | 705 | 100.00 | 1422 | 100.00 | 2384 | 100.00 |

* For this count, the texts published on the different campuses were considered part of the main institution. Colpos: Colegio de Postgraduados; Inifap: Instituto Nacional de Investigaciones Forestales, Agrícolas y Pecuarias; Uach: Universidad Autónoma Chapingo; Inecol: Instituto Nacional de Ecología; Ipn: Instituto Politécnico Nacional; Unam: Universidad Nacional Autónoma de México; Uanl: Universidad Autónoma de Nuevo León; Uv: Universidad Veracruzana; Udeg: Universidad de Guadalajara; Ujed: Universidad Juárez del Estado de Durango; Umich: Universidad Michoacana de San Nicolás de Hidalgo.

Since the year 2010, institutions such as Universidad Autónoma de Nuevo León (Uanl), Universidad Veracruzana (Uv), Universidad de Guadalajara (Udeg), Universidad Juárez del Estado de Durango (Ujed), and Universidad Michoacana de San Nicolás de Hidalgo (Umich) increased their productivity in the publication of forestry texts. According to Conacyt [28], this was because of the strengthening of their postgraduate programs in the National Registry of Quality Postgraduate Programs (Padrón Nacional de Posgrados de Calidad, PNCP). On the other hand, institutions such as Instituto Nacional de Ecología (Inecol), Instituto Politécnico Nacional (Ipn), and Universidad Nacional Autónoma de México (Unam) ceased to be models for forestry themes because they became specialized in topics associated with botany [28].

### 3.3. Spatial Distribution of the Institutions That Publish Forestry Research in Mexico

For the spatial representation of the number of articles per institution that were published from 1996 to 2019, the institution of the first author was considered (on their different campuses, according to the case). It was found that the publication of forestry research in Mexico was concentrated mostly in the center of the country in institutions such as Inifap, Colpos, Uach, Inecol, Ipn, and Unam. The western region was represented by Udeg, Umich e Inifap; the northwestern by Ujed and Inifap; and the northeastern by Uanl. However, for the southeastern region, a leading institution in the publication of forestry themes has not been found. According to Aguirre-Calderón [46], this is because the rainforest vegetation is located mostly in the southeast of Mexico, and due to tradition, forestry research has been developed in forest vegetation that is located mainly in the north of the country (Figure 4).

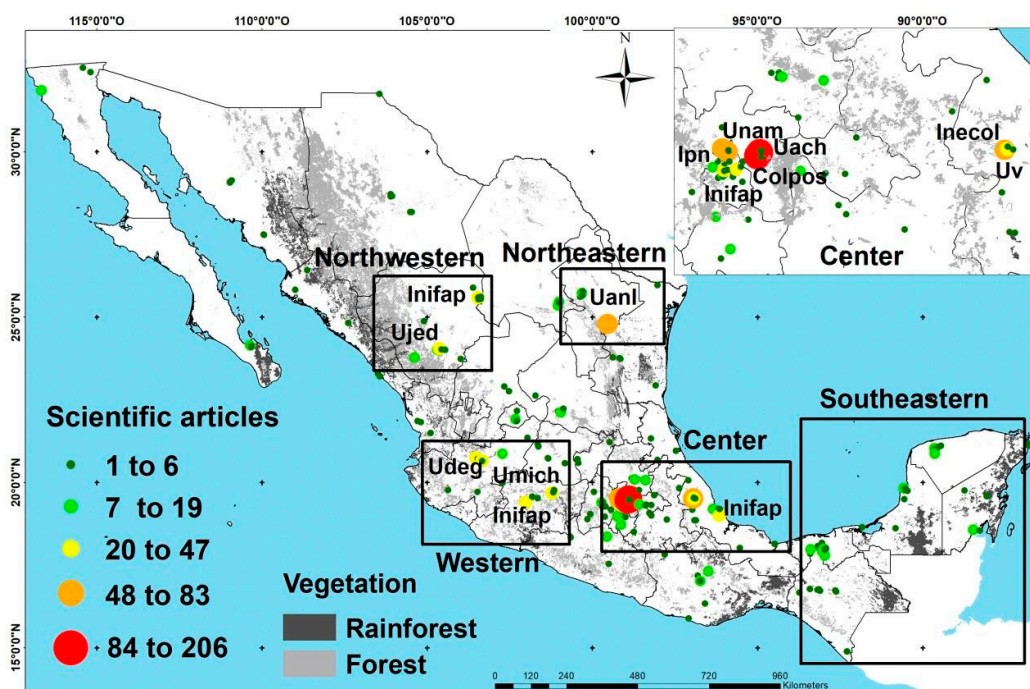

**Figure 4.** Spatial coverage of the main institutions that published scientific articles about forestry topics in Mexican journals from 1996 to 2019. Colpos: Colegio de Postgraduados; Inifap: Instituto Nacional de Investigaciones Forestales, Agrícolas y Pecuarias; Uach: Universidad Autónoma Chapingo; Inecol: Instituto Nacional de Ecología; Ipn: Instituto Politécnico Nacional; Unam: Universidad Nacional Autónoma de México; Uanl: Universidad Autónoma de Nuevo León; Uv: Universidad Veracruzana; Udeg: Universidad de Guadalajara; Ujed: Universidad Juárez del Estado de Durango; Umich: Universidad Michoacana de San Nicolás de Hidalgo.

## 4. Conclusions

The space-time evolution of scientific production on forestry published by Mexican journals presented an exponential growth from 1996 to 2019; was produced primarily by researchers from three institutions located spatially from the center to the north of the country: Instituto Nacional de Investigaciones Forestales, Agrícolas y Pecuarias (center-north), Colegio de Postgraduados (center) and Universidad Autónoma Chapingo (center). The spatial location of research institutions from the center to the north of the country coincided with areas where the forest regions in Mexico are located, which showed the need to generate more forestry research for the southeast of the country, which is where the country's rainforest regions are located.

Researchers responded to scientific trends according to the international and national themes of the moment by increasing the frequency of publications from 2011–2019 on themes associated with the sustainability of natural resources, ecology, climate change, forest biometry, carbon capture, and remote perception, and to the detriment of themes related to botany, silviculture, dasometry, and forest nursery that had the highest frequencies of publication from 1996 to 2010. In addition, a phenomenon called academic endogamy was observed, which consisted of publishing in journals of the affiliation institution of the authors, which helps explain why the journals did not reach relevant impact factors in the Journal Citation Reports. However, it should be considered that these results are valid for the analysis period (1996–2019) and correspond to publications that originated and were published in Mexico.

*Prospective Analysis*

Checking whether researchers publish in journals edited by their own institution of affiliation was the starting hypothesis of the research Bibliometric analysis of forestry

research in Mexico published by Mexican journals. This initial hypothesis constitutes the main limitation of the study since it circumscribed the search for scientific articles in Mexican journals that publish forestry issues. The bibliometric analysis of the scientific articles collected allowed for determining topics and geographical regions with potential for the development of future forestry research, which is a relevant aspect when we consider that no published research evaluated the development of the forestry sector in Mexico. However, for future related research, the authors suggest expanding the search of scientific articles in databases of international bibliographic references.

**Author Contributions:** Conceptualization, information analysis, and writing of original draft, A.S.-F.; Data review and monitoring results, N.V.-B., E.C.-Á. and J.F.J.-L.; Information analysis and writing of the final manuscript, L.M.P.-R., C.A.O.-G. and E.P.-H.; Writing, revising and editing of the final manuscript, A.E.T.-N., J.E.V.-L. and J.B.-O. All authors have read and agreed to the published version of the manuscript.

**Funding:** This research received no external funding.

**Data Availability Statement:** The data are available from the author of correspondence at reasonable request.

**Acknowledgments:** This study is part of Project number 364. Sustainable productive reconversion for the development of rural producers in Campeche, assigned to the first author by the Consejo Nacional de Ciencia y Tecnologia (CONACyT). Thanks to the anonymous reviewers of the article for their comments, which helped to enrich the research.

**Conflicts of Interest:** The authors declare no conflict of interest.

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
