# Peer review of "Bibliometric Analysis of Forestry Research in Mexico Published by Mexican Journals"

_forests, doi:10.3390/f14030648_

Round 1
Reviewer 1 Report
There are some passages that need editing to make more sense in English, e.g., lines 46-49, "The growth of scientific production in recent decades and its indexation in automatized bibliographic databases have potentialized the use of bibliometry and the generation of indicators to measure the results from scientific and technological activity."
The authors presented interesting information on the state of forestry journal article publications in Mexico.
Author Response
Dear Reviewer
The Authors reply to:
Comments and Suggestions for Authors
There are some passages that need editing to make more sense in English, e.g., lines 46-49, "The growth of scientific production in recent decades and its indexation in automatized bibliographic databases have potentialized the use of bibliometry and the generation of indicators to measure the results from scientific and technological activity."
The authors presented interesting information on the state of forestry journal article publications in Mexico.
Authors reply: Dear reviewer, the authors appreciate your comments.
Text in lines 46-49 were edited for clarity as follows: Increased scientific production in recent decades and its indexation in automatized bibliographic databases have potentiated the use of bibliometry; most likely because bibliometric analyses enable the generation indicators to measure the results from scientific and technological activities.
We include this precision in the text.

Reviewer 2 Report
The manuscript entitled ‘Bibliometric analysis of forestry research in Mexico published by Mexican journals’ by Santillán-Fernández et al. is a well written manuscript. The manuscript is worth reading and I found it different from other bibliometric analysis papers.
Abstract is well written.
Introduction is well written. However, there are some typological/grammatical errors which should be corrected. Further, introduction has been unnecessarily divided into paragraphs; better to merge these 2 – 3.
Materials and Methods describe the methodology in a scientific manner and certainly is replicable.
Results are well written and well presented. Figure quality is excellent. However, discussion can be improved.
Conclusion highlights the key finding of the manuscript.
I suggest the authors should add more information on the research gaps as well as add some recommendations in a separate sub-heading, if possible.
Overall, the manuscript is well written and I endorse it for publication.
Author Response
Dear Reviewer
The Authors reply to:
Comments and Suggestions for Authors
The manuscript entitled ‘Bibliometric analysis of forestry research in Mexico published by Mexican journals’ by Santillán-Fernández et al. is a well written manuscript. The manuscript is worth reading and I found it different from other bibliometric analysis papers.
Abstract is well written.
Introduction is well written. However, there are some typological/grammatical errors which should be corrected. Further, introduction has been unnecessarily divided into paragraphs; better to merge these 2 – 3.
Materials and Methods describe the methodology in a scientific manner and certainly is replicable.
Results are well written and well presented. Figure quality is excellent. However, discussion can be improved.
Conclusion highlights the key finding of the manuscript.
I suggest the authors should add more information on the research gaps as well as add some recommendations in a separate sub-heading, if possible.
Authors reply: Dear reviewer, the authors appreciate your comments.
Section 4.1 Prospective analysis was added: Checking whether researchers publish in journals edited by their own institution of affiliation was the starting hypothesis of the research Bibliometric analysis of forestry research in Mexico published by Mexican journals. This initial hypothesis constitutes the main limitation of the study, since it circumscribed the search for scientific articles to Mexican journals that publish forestry issues. The bibliometric analysis of the scientific articles collected allowed to determine topics and geographical regions with potential for the development of future forestry research, which is a relevant aspect when we consider that there is no published research that evaluated the development of the forestry sector in Mexico. However, for future related research, the authors suggest expanding the search of scientific articles in databases of international bibliographic references.
We include this precision in the text.

Reviewer 3 Report
REVIEW COMMENTS
I have only a few concerns about the paper and some suggestions that maybe the authors could consider:
1. To begin with, there are some typos and grammar mistakes. Some long sentences could make readers confused.
2. In the 'Introduction' section, the proposed research gap and the stated objectives do not meet the criteria of proper synergy. Please make the research gap and the research objectives consistent with each other.
3. I think the “Introduction” section can be improved by adding updated reference in paragraphs 1 and 2 in Page 2. I suggest a refs. ‘scientometric analysis of scientific literature on neuromarketing tools in advertising ', 'neuromarketing research in the last five years: a bibliometric analysis'. I think these references can help you in the issue.
4. I think the “Materials and Methodology” section can be improved by referencing several articles that have used bibliometric analysis to highlight the importance of bibliometric analysis, which has recently become widely used. I suggest some ref. 'current trends in the application of eeg in neuromarketing: a bibliometric analysis', which can be beneficial for this issue.
5. It might be appropriate for the authors to explain why they had chosen the period of extracting data between 1996 and 2019 as described in the “Materials and Methods” and Abstract sections.
6. The authors should explain why they had ignored the duration of 2020-2022.
7. The authors should clearly show the keywords that have been used in the search string to extract papers.
8. The author should explain the process of collected or extracted data clearly for replication. For example, the authors did not refer to what types of documents (e.g., article, book, book chapter, …, etc) that they have extracted from the journals to category VI of Biotechnology and Agricultural Sciences database.
9. The authors focused on all language in the paper, thereby the authors should display the number of articles and in which language.
10. I suggest the authors use the full name of abbreviations that are used in tables, as note line under each table. I think the above mentioned references can be helpful in this issue.
11. The development of search criteria does not justify why the decisions are made. So, the authors should clarify why they have chosen the journals to category VI of Biotechnology and Agricultural Sciences database for extraction papers and not WoS or Scopus database or both.
12. Could the authors explain why they have used R-tool and Ghephi software, and not VOSviewer or both?
13. The authors should explicitly state the novel contribution of this work and its similarities and differences with their previous publications.
14. The authors need to clearly articulate the academic as well as practical implications of this study in a separate section which can be named the theoretical and practical implications of this study.
15. The authors need to clearly articulate the limitations and future research of this study in a separate section which can be named ‘limitations and future research’ behind the conclusion section.
16. For readers to quickly catch your contributions, it would be better to highlight major difficulties and challenges and your original achievements to overcome them in a clearer way in the abstract and introduction.
17. How could/should your study help future studies?
If these revisions can be made to the manuscript, I believe that this study can be accepted for publication.
I wish the authors all the very best with this study.
Author Response
Dear Reviewer
The Authors reply to:
Comments and Suggestions for Authors
I think the “Introduction” section can be improved by adding updated reference in paragraphs 1 and 2 in Page 2. I suggest a refs. ‘scientometric analysis of scientific literature on neuromarketing tools in advertising ', 'neuromarketing research in the last five years: a bibliometric analysis'. I think these references can help you in the issue.
Authors reply: Dear reviewer, the authors appreciate your comments.
The following references were added:
Pilelienė, L.; Alsharif, A. H.; Alharbi, I. B. Scientometric Analysis of Scientific Literature on Neuromarketing Tools in Advertising. Baltic Journal of Economic Studies. 2022, 8, 1-12. https://doi.org/10.30525/2256-0742/2022-8-5-1-12
Alsharif, A. H.; Md Salleh, N. Z.; Baharun, R.; Rami Hashem E, A. Neuromarketing research in the last five years: A bibliometric analysis. Cogent Business & Management. 2021, 8, 1978620. https://doi.org/10.1080/23311975.2021.1978620
We include this precision in the text.
I think the “Materials and Methodology” section can be improved by referencing several articles that have used bibliometric analysis to highlight the importance of bibliometric analysis, which has recently become widely used. I suggest some ref. 'current trends in the application of eeg in neuromarketing: a bibliometric analysis', which can be beneficial for this issue.
Authors reply: Dear reviewer, the authors appreciate your comments.
The following reference was added:
Alsharif, A.; Salleh, N. Z. M.; Pilelienė, L.; Abbas, A. F.; Ali, J. Current Trends in the Application of EEG in Neuromarketing: A Bibliometric Analysis. Scientific Annals of Economics and Business. 2022, 69, 393-415. https://doi.org/10.47743/saeb-2022-0020
We include this precision in the text.
It might be appropriate for the authors to explain why they had chosen the period of extracting data between 1996 and 2019 as described in the “Materials and Methods” and Abstract sections. The authors should explain why they had ignored the duration of 2020-2022.
The authors should clearly show the keywords that have been used in the search string to extract papers.
Authors reply: Dear Reviewer, the authors appreciate the observation, in this regard this is our comment: In the first paragraph of section 2.1. Origin of the Information and data preparation of Materials and Methods, it is specified: "This study considered the journals to category VI of Biotechnology and Agricultural Sciences listed in the Classification System of Mexican Journals of the National Council of Science and Technology which publishes forestry topics, according to the guide for authors to the journals". Therefore, for the search of the scientific articles analyzed in this study, keywords were not used as a search engine for such scientific articles.
The author should explain the process of collected or extracted data clearly for replication. For example, the authors did not refer to what types of documents (e.g., article, book, book chapter, …, etc) that they have extracted from the journals to category VI of Biotechnology and Agricultural Sciences database.
Authors reply: Dear Reviewer, the authors appreciate the observation, in this regard we comment: In line 90 of section 2.1. Origin of the Information and data preparation of Materials and Methods, the sentence was supplemented: This study considered the scientific articles available in the journals to category VI of Biotechnology and Agricultural Sciences listed in the Classification System of Mexican Journals of the National Council of Science and Technology which publishes forestry topics, according to the guide for authors to the journals. We include this precision in the text.
The authors focused on all language in the paper, thereby the authors should display the number of articles and in which language.
Authors reply: Dear Reviewer, the authors appreciate your observation, in this regard we comment: In lines 123-124 we supplemented the sentence: Variables entry was done in a spreadsheet, and the language of all the texts analyzed was Spanish, which is the original language in which the analyzed journals were published. We include this precision in the text.
I suggest the authors use the full name of abbreviations that are used in tables, as note line under each table. I think the above-mentioned references can be helpful in this issue.
Authors reply: In Table 3 the full names of abbreviations were added. We include this precision in the text.
The development of search criteria does not justify why the decisions are made. So, the authors should clarify why they have chosen the journals to category VI of Biotechnology and Agricultural Sciences database for extraction papers and not WoS or Scopus database or both.
Authors reply: Dear reviewer, we appreciate this point about Master Journal List of the Web of Sciences and Scopus group. The original idea of this manuscript was to analyze all the forestry research done in Mexico which was then published in Mexican journals. This is why we didn't consider other journals that are not based in Mexico. This is was clarified in Material and Methods as follow: Only the articles whose information originated in Mexico were considered; that is, studies from foreign authors who conducted studies in Mexico were included, and studies whose study area was not located in Mexico were excluded even when published by a Mexican journal.
Could the authors explain why they have used R-tool and Ghephi software, and not VOSviewer or both?
Authors reply: Dear Reviewer, the authors appreciate the observation. The VOSviewer software was not used in this work basically due to ignorance by the authors. However, we will take into account your suggestion in future analysis.
The authors need to clearly articulate the academic as well as practical implications of this study in a separate section which can be named the theoretical and practical implications of this study.
The authors need to clearly articulate the limitations and future research of this study in a separate section which can be named ‘limitations and future research’ behind the conclusion section.
For readers to quickly catch your contributions, it would be better to highlight major difficulties and challenges and your original achievements to overcome them in a clearer way in the abstract and introduction.
How could/should your study help future studies?
Authors reply: Dear reviewer, the authors appreciate your comments.
Section 4.1 Prospective analysis was added: Checking whether researchers publish in journals edited by their own institution of affiliation was the starting hypothesis of the research Bibliometric analysis of forestry research in Mexico published by Mexican journals. This initial hypothesis constitutes the main limitation of the study, since it circumscribed the search for scientific articles to Mexican journals that publish forestry issues. The bibliometric analysis of the scientific articles collected allowed to determine topics and geographical regions with potential for the development of future forestry research, which is a relevant aspect when we consider that there is no published research that evaluated the development of the forestry sector in Mexico. However, for future related research, the authors suggest expanding the search of scientific articles in databases of international bibliographic references.
We include this precision in the text.

Round 2
Reviewer 3 Report
REVIEW COMMENTS
I have only a few concerns about the paper and some suggestions that maybe the authors could consider:
1. To begin with, there are some typos and grammar mistakes. Some long sentences could make readers confused.
2. The authors should explicitly state the novel contribution of this work and its similarities and differences with their previous publications.
3. The authors need to clearly articulate the academic as well as practical implications of this study in a separate section which can be named the theoretical and practical implications of this study.
4. The authors need to clearly articulate the limitations and future research of this study in a separate section which can be named ‘limitations and future research’ behind the conclusion section.
5. For readers to quickly catch your contributions, it would be better to highlight major difficulties and challenges and your original achievements to overcome them in a clearer way in the abstract and introduction.
I wish the authors all the very best with this study.
Author Response
Response to Reviewer 3 Comments
Round 2
Dear Reviewer
The Authors reply to:
Comments and Suggestions for Authors
- To begin with, there are some typos and grammar mistakes. Some long sentences could make readers confused.
Authors reply: Dear reviewer, the authors appreciate your comments.
The original language of the work was Spanish. However, the translation was sent to experts on the subject whose native language is English. The translation has been well reviewed, however, the authors commit ourselves, in case they consider it so, to use the English language editing services available on the MDPI web portal.
- The authors should explicitly state the novel contribution of this work and its similarities and differences with their previous publications.
Authors reply: Dear Reviewer, the authors appreciate your observation, in this regard we comment: Section 4.1 Prospective analysis was added. In lines 346-349 it is specified: The bibliometric analysis of the scientific articles collected allowed to determine topics and geographical regions with potential for the development of future forestry research, which is a relevant aspect when we consider that there is no published research that evaluated the development of the forestry sector in Mexico.
In addition, in the Abstract, lines 33-37 we supplemented the sentence: Also showed the need to generate more forestry research for the southeast of the country, in topics such as climate change, carbon capture, forest biometry, and remote perception. We include this precision in the text.
- The authors need to clearly articulate the academic as well as practical implications of this study in a separate section which can be named the theoretical and practical implications of this study.
Authors reply: Dear reviewer, the authors appreciate your comments.
Section 4.1 Prospective analysis was added: Checking whether researchers publish in journals edited by their own institution of affiliation was the starting hypothesis of the research Bibliometric analysis of forestry research in Mexico published by Mexican journals. This initial hypothesis constitutes the main limitation of the study, since it circumscribed the search for scientific articles to Mexican journals that publish forestry issues. The bibliometric analysis of the scientific articles collected allowed to determine topics and geographical regions with potential for the development of future forestry research, which is a relevant aspect when we consider that there is no published research that evaluated the development of the forestry sector in Mexico. However, for future related research, the authors suggest expanding the search of scientific articles in databases of international bibliographic references.
- The authors need to clearly articulate the limitations and future research of this study in a separate section which can be named ‘limitations and future research’ behind the conclusion section.
Authors reply: Dear Reviewer, the authors appreciate your observation, in this regard we comment: Section 4.1 Prospective analysis was added.
In lines 342-346 it is specified: Checking whether researchers publish in journals edited by their own institution of affiliation was the starting hypothesis of the research Bibliometric analysis of forestry research in Mexico published by Mexican journals. This initial hypothesis constitutes the main limitation of the study, since it circumscribed the search for scientific articles to Mexican journals that publish forestry issues.
In lines 350-351 it is specified: However, for future related research, the authors suggest expanding the search of scientific articles in databases of international bibliographic references.
- For readers to quickly catch your contributions, it would be better to highlight major difficulties and challenges and your original achievements to overcome them in a clearer way in the abstract and introduction.
Authors reply: Dear reviewer, the authors appreciate your comments.
In lines 33-37 of the Abstract section, we supplemented the sentence: Also showed the need to generate more forestry research for the southeast of the country, in topics such as climate change, carbon capture, forest biometry, and remote perception; which are relevant aspects when we consider that there is no published research that evaluated the development of the forestry sector in Mexico. We include this precision in the text.
